# Bactericidal Effect and Associated Properties of Non-Electrolytic Hypochlorite Water on Foodborne Pathogenic Bacteria

**DOI:** 10.3390/foods11244071

**Published:** 2022-12-16

**Authors:** Xiangyu Gao, Xueqiang Liu, Jialin He, Hanbing Huang, Xiaoya Qi, Jianxiong Hao

**Affiliations:** College of Food Science and Biology, Hebei University of Science and Technology, No.26 Yuxiang Street, Shijiazhuang 050018, China

**Keywords:** non-electrolytic hypochlorite water, slightly acidic electrolytic water, disinfection, UV scanning, hydroxyl radicals, storage stability

## Abstract

This study investigated the broad-spectrum bactericidal activity of non-electrolytic hypochlorite water (NEHW) and detected its hydroxyl radical content compared with that of slightly acidic electrolytic water (SAEW). Based on the results of UV scanning and storage stability, higher hypochlorite content and stronger oxidation were found to be responsible for the stronger bactericidal effect of NEHW. NEHW can achieve 99% bacterial disinfection effect by treating with 10 mg/L available chlorine concentration for more than 5 minutes. At the same time, the storage stability of NEHW was higher than that of SAEW. After 20 days of storage under sealed and dark conditions, the pH value only increased by 7.9%, and the effective chlorine concentration remained nearly 80%. The results showed that NEHW had higher germicidal efficacy and storage stability than SAEW.

## 1. Introduction

A part of biological pollution caused by foodborne bacteria pathogens has attracted increasing attention of consumers and scientists, and has become an important issue endangering food safety [1]. Particularly in the process of food storage and transportation, contamination of bacteria pathogens, such as *Listeria monocytogenes* will form a biofilm on the pipeline [2]. Most of these bacteria are highly tolerant to environmental stress factors, and some even form biofilms [3]. They attach to the food surface and enter the human body through ingestion, causing diseases such as meningitis [3], food poisoning, and bacteremia [4].

To reduce the threat of foodborne pathogens, various methods are used in food storage and processing, including high hydrostatic pressure processing, ultrasounds, non-thermal atmospheric plasma, pulsed electric fields, electrolyzed water, and plasma-activated water [5,6,7,8,9,10]. However, the increased focus of people on food safety, the relatively low sterilization rate, and the impact of treatment on food quality have limited the application of these methods. Slightly acidic electrolytic water (SAEW) is less corrosive than other chemical disinfectants. It can effectively eliminate pathogens from fresh vegetables, fruits, seafood [9], and poultry [11]. However, SAEW development is limited because of its high dependence on production equipment. Non-electrolytic hypochlorite water (NEHW) is an aqueous solution containing hypochlorous acid as the main bactericidal component. NEHW is characterized by a rapid preparation time, broad-spectrum and efficient sterilization, high safety, harmlessness to humans, and green environmental protection [12]. NEHW has an evident sterilization effect and is reduced to an aqueous solution after the complete reaction. It does not produce highly toxic disinfection by-products and can maintain its safety characteristics during sterilization [13]. Its main substance is the same as hypochlorite produced in human body. The cells use mitochondrial binding enzymes to convert O_2_ into H_2_O_2_, and then, myeloperoxidase from neutrophils catalyzes the reaction between H_2_O_2_ and Cl^-^ to produce HClO, which plays a bactericidal role in humans [14]. Yu and others reported that NEHW cleared the acute oral toxicity, micronucleus, and acute inhalation tests according to the technical specifications for disinfection issued by the Ministry of Health of China [15]. The test results showed that pH was 6.5, and the acute oral LD_50_ of 50 ppm hypochlorous acid water in male and female mice at 25 °C was greater than 5000 mg/kg. The micronucleus test of polychromatic erythrocytes in the bone marrow of mice in hypochlorite water group was negative. The acute inhalation toxicity LC_50_ (2 h) of hypochlorite water to male and female mice was greater than 10,000 mg/m^3^, which indicates that NEHW is non-toxic [15].

NaClO is dissolved in water to produce sodium hypochlorite solution, which contains HClO. HClO oxidation is the main factor responsible for bacterial inactivation [16]. Studies have shown that the HClO concentration in NaClO solution changes with a change in pH [17]. When the pH of the solution ranged from 2–3, the existing form of effective chlorine was mostly Cl_2_, and when the pH was close to alkaline, the existing form was ClO^-^. When the pH ranged from 5.5–6.5, the existing form was HClO [18]. The bactericidal activity of Cl_2_ and ClO^-^ is weaker than that of HClO. Therefore, NEHW with a high bactericidal activity can be obtained by adjusting the pH value of the sodium hypochlorite solution. Compared with acid electrolytic water (AEW, pH 2.3–2.8, available chlorine concentration 60–200 mg/L) and SAEW (pH 5.5–6.5, available chlorine concentration 10–30 mg/L), NEHW showed strong corrosivity, high chlorine residue content, great dependence on equipment, and a complex preparation method [19,20,21]. However, most current studies have only proven that NEHW has a certain bactericidal activity and has a low production cost [22]. NEHW has not been compared with the germicidal efficacy of other existing disinfectants, and relevant studies on the germicidal mechanism, pertinent physical and chemical indicators, and storage stability of NEHW are lacking.

In this experiment, the bactericidal activity of NEHW and SAEW under the same pH available chlorine and bactericidal time treatment was compared. The factors causing the difference of bactericidal activity between NEHW and SAEW were studied by ultraviolet scanning of hypochlorite content of the two biocides under the same pH and available chlorine conditions and detection of hydroxyl radical. Finally, the research on storage stability further shows the advantages of NEHW in the production process.

## 2. Materials and Methods

### 2.1. Bacterial Strain

*Listeria monocytogenes* ATCC19114 was purchased from Beijing Solarbio Biotechnology Co., Ltd. (Beijing, China) *Escherichia coli* O78, used in the experiments, was obtained from Institute of Microbiology, Chinese Academy of Sciences (Beijing, China). The standard strain of *Staphylococcus aureus* ATCC6538 was provided by Guangdong Microbial Strain Collection Center (Guangzhou, China). The *Candida albicans* strain AY93025 was purchased from the China Culture Preservation Center (Beijing, China). *Bacillus subtilis* CS27 was isolated and preserved by the Institute of Soil and Fertilizer, Fujian Academy of Agricultural Sciences (Fuzhou, China), and identified by the Institute of Microbiology, Chinese Academy of Sciences (Beijing, China).

In brief, all operations were performed according to the Technical Standard for Disinfection.

*L. monocytogenes* was cultured in 200 mL TSB-YE at 37 °C for 24 h with agitation (160 rpm). *E. coli* was cultured in 200 mL nutrient broth at 37 °C for 24 h with agitation (170 rpm). *S. aureus* was cultured in 200 mL Sabouraud glucose liquid medium at 37 °C for 24 h with agitation (155 rpm). *C. albicans* was cultured in 200 mL nutrient broth at 37 °C for 24 h with agitation (160 rpm), and *B. subtilis* was cultured in 200 mL Sabouraud glucose liquid medium at 28 °C for 24 h with agitation (155 rpm). After cultivation, 1 mL of each culture was harvested by centrifugation for 10 min at 8000× *g* (Model TGL-16G, Shanghai ANTING Scientific Instrument Factory, Shanghai, China). The pellets were washed twice with 1 mL normal saline and resuspended in normal saline to obtain the final cell concentration of approximately 10^9^ cfu/mL. The dilution coating plate method was used to estimate the bacterial solution concentration.

### 2.2. Preparation of Treatment Solutions

SAEW was prepared using a flow type electrolysis apparatus (non-membrane electrolytic cell, model AQUACIDO NDX-250KMS, OSG Company Ltd., Shinshiro, Japan), and NEHW was made from sodium hypochlorite solution by adjusting pH and diluting it with water. After preparation, the two types of water were stored in polypropylene containers, and immediately used for measurement. The pH values of these two water types were measured using a pH meter (Multifunctional laboratory pH meter, model ST2100, OHAUS Instrument Co., Ltd., Shanghai, China), and their available chlorine concentration (ACC) values were measured using the iodometric method. ACC was calculated using the following equation (the numbers 35.45 and 10 are according to the manufacturer):ACC (mg/L) = (V × M × 35.45) × 10(1)

V is the consumption of sodium thiosulfate (mL),M is the concentration of sodium thiosulfate (mol/L).

To evaluate the germicidal efficacy of NEHW and SAEW with different ACC values, these two water types having different dilution times were used in this experiment. The tap water was used as a control. The pH and ACC of all the treatment solutions are shown in Table 1.

### 2.3. NEHW and SAEW Treatments

According to the method of [23], 9 mL of NEHW and SAEW were transferred to separate test tubes containing 1 mL of 10^9^ cfu/mL strain inoculum. After treatment with 10 and 30 effective chlorine concentration for 0.5 min, 1 min, 5 min, and 10 min, respectively, the reaction was stopped by adding 9 mL neutralizer (mixed with 7.14 g/L disodium hydrogen phosphate, 1.36 g/L potassium dihydrogen phosphate, and 1.56 g/L sodium thiosulfate) to 1 mL test suspension to eliminate the residual activities of NEHW and SAEW [24]. The results of sterilization will be described later in this paper.

### 2.4. Bacterial Enumeration

The spread plate method was used for bacterial enumeration. A total of 0.1 mL bacterial solution was first placed on the corresponding solid culture medium, and then applied evenly with the triangular plating rod [25,26]. *L. monocytogenes* was spread-plated onto TSB-YE solid medium. *E. coli* was spread-plated onto nutrient agar. *S. aureus* was spread-plated onto Sabouraud glucose agar medium. *C. albicans* was spread-plated onto nutrient agar. *B. subtilis* was spread-plated onto Sabouraud glucose agar medium. The plates were incubated at 37 °C for 24–48 h, colonies were counted, and the number of viable bacteria was expressed in lg cfu/mL.

### 2.5. UV Scanning of NEHW and SAEW

To explore the reasons for the different bactericidal effects of NEHW and SAEW, UV scanning was conducted using a spectrophotometer (Model UV-5200, YUANXI Instrument Co., Ltd., Shanghai, China) in the range of 220–380 nm [27]. A scan within the ultraviolet absorption wavelength of the existing form of the common effective chlorine present in the known disinfectant was used to detect the existing form and the difference in the effective chlorine content of the two disinfectants.

### 2.6. Hydroxyl Radical Content Detection

Hydroxyl radical content of the biocide was measured through UV spectrophotometry [28]. The detection reaction was designed based on the Fenton reaction. The change in the absorbance value, that is, the hydroxyl radical content, before and after the reaction was detected by dyeing with a dye, and compared. The number of free radicals in the system was determined through spectrophotometry after being dyed with salicylic acid [29], methylene blue [30], and crystal violet [31] and standing for 20 min. The absorbance of salicylic acid was measured at 660 nm, and that of methylene blue and crystal violet was measured at 530 nm. Scanning was performed using a spectrophotometer (Model UV-5200, Shanghai YUANXI Instrument Co., Ltd., Shanghai, China).

### 2.7. Storage Stability

We referred to previous studies for the methodology to be used for studying storage stability [32,33]. NEHW and SAEW adjusted to pH 5.7 and having an effective chlorine concentration of 36 were used for the storage experiment. The experiment was carried out for 20 days under the conditions of sealed and avoiding light, for which a brown bottle with a sealing cap and a transparent open glass conical bottle were used, respectively. The changes in pH and ACC of the two types of water were measured during the experiment. pH was measured using the pH meter, and the effective chlorine concentration was measured through iodometry. To ensure the comparability of results, each test was conducted in the same laboratory, and the sealed samples were no longer used after being opened. The pH values and effective chlorine concentrations of all samples were measured three times a day, and the average value was considered. Each measurement was completed within 30 min.

### 2.8. Statistical Analysis

Each treatment was repeated three times. For each treatment, data from independent replicate trials were pooled and the means and standard deviations were calculated. All data were analyzed using Duncan’s multiple range test (SPSS16.0 for Windows, SPSS Inc., Chicago, IL, USA). Significant differences between treatments were established at a significance level of *p* < 0.05.

## 3. Results and Discussion

### 3.1. Broad Spectrum Bactericidal Activity of NEHW and SAEW

The physical and chemical parameters of NEHW and SAEW used in this study are shown in Table 1. Deionized water was used to dilute NEHW and SAEW into two effective chlorine concentrations, and four sterilization times were used to sterilize five bacteria and compare the sterilization effects of the two biocides (Figure 1).

The populations of *L. monocytogenes*, *E. coli*, *S. aureus*, *C. albicans*, and *B. subtilis* control groups treated with tap water were 9.25, 8.93, 9.05, 9.00, and 8.75 cfu/mL.

At 0.5 and 1 min of treatment, the biocides with two effective chlorine concentrations significantly reduced the number of five bacteria, but did not reach the 99% bactericidal rate. After 5 min of treatment, under 10 ACC, NEHW made *L. monocytogenes*, *E. coli*, *S. aureus*, *C. albicans,* and *B. subtilis* reduced by 1.99, 1.91, 2.21, 2.23, and 2.56 lg cfu/mL, respectively (Figure 1, *p* < 0.05). SAEW reduced *L. monocytogenes*, *E. coli*, *S. aureus*, *C. albicans,* and *B. subtilis* by 1.43, 1.71, 1.97, 2.01, and 2.26 lg cfu/mL, respectively (Figure 1, *p* < 0.05). Under 30 ACC, NEHW made *L. monocytogenes*, *E. coli*, *S. aureus*, *C. albicans,* and *B. subtilis* reduced by 2.09, 1.95, 2.33, 2.36, and 2.76 lg cfu/mL, respectively (Figure 1, *p* < 0.05). SAEW made *L. monocytogenes*, *E. coli*, *S. aureus*, *C. albicans,* and *B. subtilis* reduced by 1.48, 2.03, 2.20, 2.04, and 2.50 lg cfu/mL, respectively (Figure 1, *p* < 0.05). After 10 min of treatment, under 10 ACC, NEHW made *L. monocytogenes*, *E. coli*, *S. aureus*, *C. albicans,* and *B. subtilis* reduced by 2.15, 2.29, 2.35, 2.41, and 2.71 lg cfu/mL, respectively (Figure 1, *p* < 0.05). SAEW reduced *L. monocytogenes*, *E. coli*, *S. aureus*, *C. albicans,* and *B. subtilis* by 1.49, 1.91, 2.10, 2.30, and 2.63 lg cfu/mL, respectively (Figure 1, *p* < 0.05). Under 30 ACC, NEHW made *L. monocytogenes*, *E. coli*, *S. aureus*, *C. albicans,* and *B. subtilis* reduced by 2.24, 2.43, 2.56, 2.59, and 2.98 lg cfu/mL, respectively (Figure 1, *p* < 0.05). SAEW reduced *L. monocytogenes*, *E. coli*, *S. aureus*, *C. albicans,* and *B. subtilis* by 1.54, 2.18, 2.32, 2.36, and 2.76 lg cfu/mL, respectively (Figure 1, *p* < 0.05).

For *L. monocytogenes*, NEHW could achieve a 99.28% bactericidal rate after 10 min of treatment at 10 ACC, while the bactericidal rate of SAEW reached 97.37% after 10 min of treatment under 30 ACC. For *E. coli*, NEHW could achieve a 99.46% bactericidal rate after 10 min of treatment at 10 ACC, while the bactericidal rate of SAEW reached 99.31% after 10 min of treatment under 30 ACC. For *S. aureus*, NEHW could achieve a 99.36% bactericidal rate after 5 min of treatment at 10 ACC, while the bactericidal rate of SAEW reached 99.34% after 10 min of treatment under 30 ACC. For *C. albicans*, NEHW could achieve a 99.37% bactericidal rate after 5 min of treatment at 10 ACC, while the bactericidal rate of SAEW reached 99.46% after 10 min of treatment under 10 ACC. For *B. subtilis*, NEHW could achieve a 99.10% bactericidal rate after 5 mins of treatment at 30 ACC, while the bactericidal rate of SAEW reached 99.07% after 10 min of treatment under 30 ACC.

Under the conditions, both disinfectants could reduce the strain concentration by at least 1.5 lg cfu/mL. NEHW achieved the 99% killing rate after 15 min of sterilization at 10 ACC, while SAEW achieved the same effect after 5 min of sterilization at 30 ACC or 10 min of sterilization at 10 ACC. For the same strain, the difference in the bactericidal effect of NEHW at 30 and 10 ACC was greater than that of SAEW. The sterilization effect of NEHW was significantly higher than that of SAEW, which was not above the limitation < 3. Studies have focused on the bactericidal activity of SAEW against various bacteria, including *B. cereus*, *Vibrio vulnificus,* and *V. parahaemolyticus*, *Salmonella enteritidis,* and *Aspergillus flavus* [17,32,34,35]. Our results are in agreement with those of the previous report. For the previous similar studies, NEHW and plasma-activated water both showed good germicidal efficacy. At the same time, plasma-activated water has also proven that it can be used to inactivate biofilm-forming pathogens from stainless steel surfaces [10].

Although the pH did not change significantly after different dilution degrees of NEHW and SAEW, significant differences were observed in the germicidal efficacy of the two biocides at the same ACC. Previous studies considered ACC as the main disinfection factor [36,37]. However, in this study, almost the same ACC showed significantly different bactericidal effects of NEHW and SAEW. For example, NEHW at 30 ACC reduced *L. monocytogenes* by 2.24 lg cfu/mL (over the limitation < 3), but SAEW at the same ACC only decreased the bacteria by 1.59 lg cfu/mL. This may indicate that effective chlorine of similar concentrations may also comprise of different forms of effective chlorine, resulting in different bactericidal capacities.

### 3.2. Scanning Results of Effective Chlorine Substances

Scanning was conducted in the wavelength range of 220–380 nm, including HClO with the maximum absorption peak at 234 nm with molar absorptivity of 100 cm^−1^ M^−1^ and ClO^-^ with the maximum absorption peak at 292 nm with molar absorptivity 350 cm^−1^ M^−1^. This allows us to detect and analyze the difference in the existing form and content of effective chlorine of two biocides with the same effective chlorine and pH. Scanning results showed that both NEHW and SAEW had maximum absorption at 234 nm, as shown in Figure 2. The results are consistent with those of the previously reported theoretical methods [27,38]. The two biocides were scanned at 10 and 30 ACC, respectively. The absorption peak at 234 nm for NEHW was significantly higher than that for SAEW (*p* < 0.05). Hypochloric acid concentration can be calculated by Beer’s law equation, where A is absorbance, K is molar absorptivity (M^−1^ cm^−1^), b is thickness of absorption layer (cm), and c (mol/L) is concentration of HClO. According to the formula, the K value and b value of NEHW and SAEW are the same, so the absorption value A can directly reflect the hypochlorite concentration c level of NEHW and SAEW. Therefore, it can be determined that the HClO concentration of NEHW is higher than that of SAEW.

Previous studies have found that pH plays crucial role in determining the form of available chlorine in electrolyzed water biocides, including hypochlorite (HClO), hypochlorite ion (ClO^-^), and chlorine (Cl_2_). When the pH value ranged from 5.5–6.5, the main form of effective chlorine in the biocide was HClO. At the same concentration, the sterilization effect of hypochlorous acid on bacteria was dozens of times of that of the hypochlorite ion (ClO^-^) [39,40]. Under alkaline conditions, the main existing effective chlorine form in the sodium hypochlorite disinfectant was ClO^-^, and that in the acidic electrolytic water with pH of 2.0–2.5, was dissolved chlorine Cl_2_. Because of the loss caused by its easy volatilization [33], the bactericidal activity of ClO^-^ was not as good as that of HClO, which also makes NEHW stronger than SAEW, NaClO, and AEW.

### 3.3. Hydroxyl Radical Detection of NEHW and SAEW

The ability of NEHW and SAEW to produce hydroxyl radicals was tested through the Fenton reaction [41,42]. The physical and chemical indices of NEHW and SAEW used are presented in Table 1. In this experiment, we evaluated the hydroxyl radical content of different types of NEHW and SAEW. After color development with three reagents, the results were obtained through UV scanning, which detected the loss of salicylic acid, crystal violet, and methylene blue [28,43]. The results are shown in Figure 3. At 10 and 30 ACC, the hydroxyl radical absorption of NEHW was 0.329 and 0.377, respectively, higher than that of SAEW at 0.304 and 0.342, respectively. During the detection of salicylic acid, the hydroxyl radical absorption of NEHW at 10 and 30 ACC was 0.148 and 0.195, which is higher than that of SAEW at 0.132 and 0.157, respectively. Under the detection of methylene blue, the hydroxyl radical absorption of NEHW at 10 ACC and 30 ACC reached 0.317 and 0.353, higher than that of SAEW at 0.312 and 0.342, respectively. At the same effective chlorine concentration and pH, the hydroxyl radical content of NEHW was always significantly higher than that of SAEW.

Hydroxyl free radicals have a strong oxidation ability [44]. They can react with proteins, lipids, and nucleic acids. Moreover, hydroxyl free radicals can locate mitochondria and cause mitochondrial dysfunction [45]. They are more detrimental to bacteria, and lead to cell destruction and death [39,40]. Compared to AEW, SAEW was recognized as a highly effective biocide in the food industry because of its lower ACC concentration and lower chlorine residue (Cl_2_) after sterilization [46]. NEHW exhibited a higher bactericidal effect than SAEW at the same physical and chemical indicies. Because of its higher content of HClO and hydroxyl radicals at the same ACC compared with other chlorinated compounds, NEHW could easily penetrate the cell wall, irreversibly oxidize key cell components, and quickly eliminate pathogens [47]. According to the previous UV wavelength scanning experiment, at the same effective chlorine, NEHW exhibited a higher bactericidal effect than SAEW because it had a higher content of HClO and hydroxyl radicals. Simultaneously, UV scanning also revealed that the effective chlorine concentration was not the only key factor affecting the disinfection effect. At the same effective chlorine concentration, the contents of HClO and hydroxyl radicals in NEHW were significantly higher than those in SAEW. Thus, NEHW could achieve a higher disinfection effect at a lower effective chlorine concentration and shorter sterilization time.

### 3.4. Storage Stability Results of NEHW and SAEW

The pH of NEHW on configuration day was 5.7, and the effective chlorine concentration was 35.5. The pH of SAEW on configuration day was 5.7, and the effective chlorine concentration was 36. The changes in pH and effective chlorine concentration of the two types of water stored under different storage conditions for 20 days are presented in Figure 4. Over different storage times, the pH of the two biocides increased and the effective chlorine concentration decreased with time. Different storage conditions have a significant impact on the change trends [46].

Under the condition of open without avoiding light, on the 7th day, the pH value of SAEW increased to 7 (Figure 4, *p* < 0.05). On the 15th day, the pH of NEHW increased to 7.14 (Figure 4, *p* < 0.05). The form of available chlorine changed from HClO to ClO^-^, and the bactericidal activity decreased significantly.

Under the condition of sealed and avoiding light, on the 7th day of treatment, the pH of SAEW increased from 5.7 to 6.46 (12.3% changed), and ACC decreased from 36 to 31.9 (11.3% changed) (Figure 4, *p* < 0.05), while the pH of NEHW increased from 5.7 to 5.92 (3.8% changed), and ACC decreased from 35.5 to 33.68 (5.1% changed) (Figure 4, *p* < 0.05). On the 20th day of treatment, the pH value of SAEW increased from 5.7 to 7.02 (23.1% changed), and ACC decreased from 36 to 24.80 (31.1% changed) (Figure 4, *p* < 0.05), while the pH value of NEHW increased from 5.7 to 6.15 (7.9% changed), and ACC decreased from 35.5 to 28.36 (20.1% changed) (Figure 4, *p* < 0.05).

Unlike AEW, the effective chlorine of NEHW and SAEW was not Cl_2_, which was strongly volatile. However, when hypochlorous acid was exposed to light and air, light accelerated the decomposition of hypochlorous acid into hydrochloric acid and oxygen, and exposure to air further volatilized these substances. This led to a rise of in the pH of the biocide, a decrease in its effective chlorine concentration, and finally the loss of bactericidal activity. At almost the same pH and effective chlorine concentration, NEHW displayed a higher storage stability than SAEW, volatilized more slowly under the open without avoiding light condition, and could maintain approximately stable physical and chemical indices within 20 days under the sealed and avoiding light condition.

## 4. Conclusions

In this study, NEHW and SAEW were compared for their sterilization effect, effective chlorine content, hydroxyl radical content, and storage stability. The results showed that NEHW has a broad-spectrum bactericidal activity, and under the bactericidal conditions used in this study, NEHW achieved a 99% bactericidal rate after treatment for at least 5 min when the ACC was 10. However, bacteria in the VBNC state were not considered in this experiment. At the same pH, the contents of hypochlorite, HClO, and hydroxyl radicals in the available chlorine were significantly higher in NEHW than in SAEW. Regarding storage stability, NEHW could maintain relatively stable physical and chemical indices after 10 days in open storage and 20 days in closed storage. At present, the main indicators of electrolytic water are pH, available chlorine, and ORP. AEW is volatile and highly corrosive because of its low pH, which leads to the existence of chlorine as effective chlorine. Under the same conditions as SAEW, NEHW exhibited the characteristics of higher hydroxyl radical and HClO contents, and the NEHW is less expensive than the SAEW. For food manufacturers, NEHW does not need electrolysis equipment for preparation, but only needs pH adjustment and water dilution of sodium hypochlorite solution, which saves the cost of production equipment and power.

The stability of NEHW in production, packaging, storage, transportation, and use is affected by various factors. However, the stability research results under the experimental conditions have limited guiding significance for practical application. This experiment only proved that NEHW is more stable than SAEW, and the influence of exposure and light on the stability of these biocides requires to be explored further. At the same time, combined with other research results, our finding related to storage stability proves the advantages of NEHW. Some data show that the stability of biocides is also affected by the conditions of loading, temperature, and vibration. Hypochlorous acid decomposes at room temperature, and the higher the temperature, the faster the decomposition. Excessive vibration of the biocide not sealed in a full bottle will makes the biocide to more closely come in contact with the air, resulting in decomposition. With the extension of observation time, the experimental results may change. To ensure the quality of NEHW, we recommend filling NEHW in containers up to the brim during production and choosing colored containers to minimize vibration during transportation.

## Figures and Tables

**Figure 1 foods-11-04071-f001:**
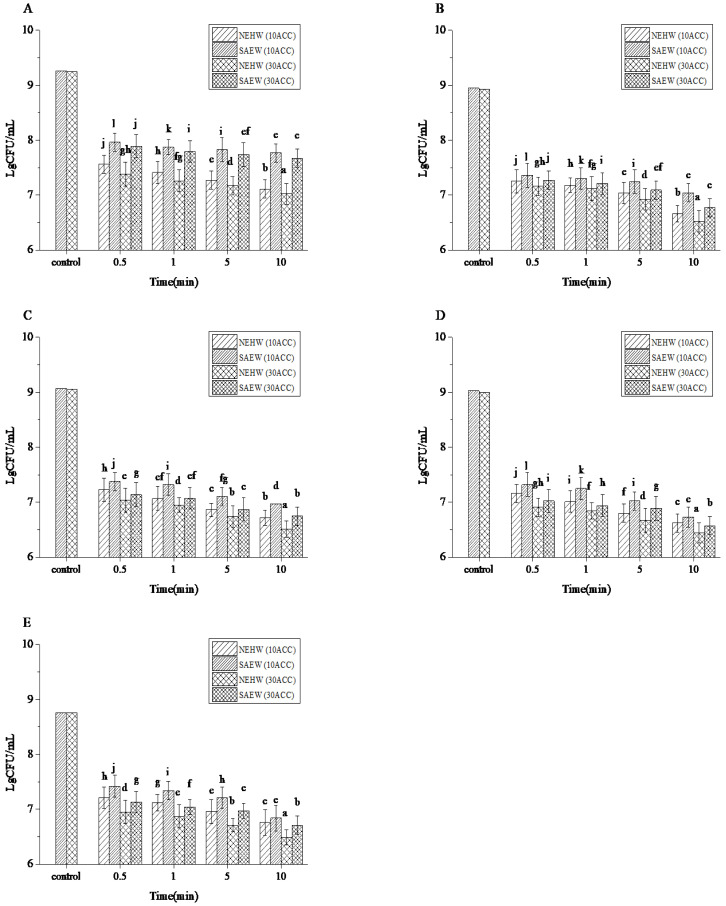
The number of remaining colonies of five bacteria after treatment with different concentrations of biocide and sterilization time, expressed by logarithm. (**A**) *Listeria monocytogenes*; (**B**) *Escherichia coli*; (**C**) *Staphylococcus aureus*; (**D**) *Candida albicans*; (**E**) *Bacillus subtilis*. All treatments and determinations were performed in triplicate. The different letters indicate significant differences (*p* < 0.05). Values are the means of three replicated measurements ± standard deviation (SD).

**Figure 2 foods-11-04071-f002:**
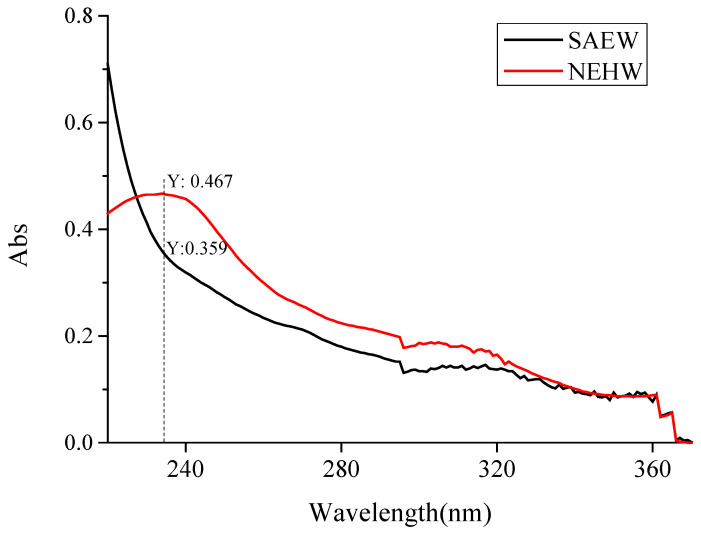
The results of NEHW and SAEW scanning at the wavelength of 220 nm—370 nm on the UV-Vis spectrophotometer. The physical and chemical parameters of NEHW and SAEW are both pH: 5.7; ACC: 30.

**Figure 3 foods-11-04071-f003:**
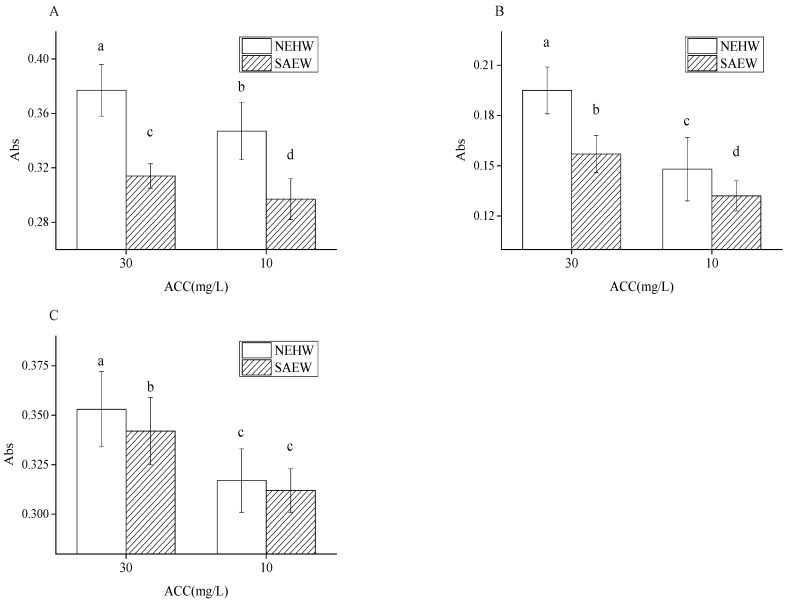
The detection of hydroxyl radical content by NEHW and SAEW under 30 and 10 ACC. (**A**) is the crystal violet detection method, (**B**) is the salicylic acid detection method, (**C**) is the methylene blue detection method. The different letters indicate significant differences (*p* < 0.05).

**Figure 4 foods-11-04071-f004:**
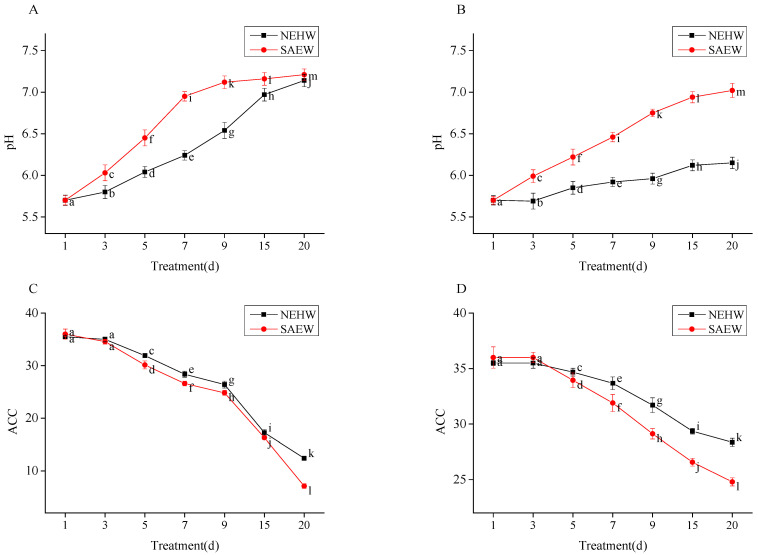
The changes of pH and ACC of NEHW and SAEW during 20 days of storage in open containers without avoiding light and sealed while avoiding light. (**A**,**C**) are the changes of pH and ACC in open containers without avoiding light, respectively; (**B**,**D**) are the changes of pH and ACC under the condition of sealed and avoiding light, respectively. The NEHW initial pH is 5.7 and ACC is 35.5; the SAEW initial pH is 5.7 and ACC is 36. ACC is the abbreviation for available chlorine concentration. All treatments and determinations were performed in triplicate. The different letters indicate significant differences (*p* < 0.05). Values are the means of three replicated measurements ± standard deviation (SD).

**Table 1 foods-11-04071-t001:** ^a,b^ Chemical parameters of different solutions.

Solutions	pH	ACC (mg/L)
SAEW1	5.67 ± 0.04	10.63 ± 0.54
SAEW2	5.69 ± 0.10	30.13 ± 0.21
NEHW1	5.66 ± 0.06	10.98 ± 0.35
NEHW2	5.70 ± 0.12	30.84 ± 0.20
Tap water (TW)	7.79 ± 0.04	ND

^a^ SAEW was the abbreviation of slightly acidic electrolyzed water and NEHW was the abbreviation of non electrolytic hypochlorite water. ACC was the abbreviation of available chlorine concentration. Tap water as control was the drinking water that came from Hebei University of Science and Technology. ^b^ Data are expressed by mean ± standard deviation (SD) and values were obtained by three replicated measurements.

## Data Availability

Data is contained within the article.

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
