# Peer review of "Bactericidal Effect and Associated Properties of Non-Electrolytic Hypochlorite Water on Foodborne Pathogenic Bacteria"

_foods, 2022, doi:10.3390/foods11244071_

Round 1
Reviewer 1 Report
Line 93: Ministry of health, where? China? If yes, add ''China'' after 'Health'
Lines 105-106. revise to make clearer. Could 'it' after 'diluting' be replaced with 'water'
Line 130-131: State the concentrations and times used
Line 132: What quantity of the neutraliser was added?
Line: 172: Do the authors mean 'each' instead of 'whole'
Line 235: .......two fungicides? Tests were carried out against bacteria and fungi. Biocide maybe.
Figure 2: It is not clear that the absorption peak at 234 was significantly (any p value?) higher for NEHW. Is there any way of putting a dotted line to show where peak 234 is?
What is the broad peak at 240 nm which is clearly missing in NEHW?
Conclusions: The claim for significant content of HCIO appears to be supposition. Why was a solution or standard with known HCIO not used? This needs to be carried out but if it is not possible to do this, then the conclusion needs to de revised to state that HCIO 'may' be higher and not 'significantly higher'.
Cost benefits: The food manufacturer will like to know if NEHW is less expensive.
Reviewer 2 Report
The article as a whole has been properly designed. However, I found that the Authors used viable bacterial strains in their tests and did not consoderate (I believe) to also use nonvital forms of the same bacteria. I have reported the problem in the pdf that I am attaching to these observations.
I also found some typing errors and a picture is missing in the text. I invite the authors to consider my observations.

Reviewer 3 Report
The paper describes the bactericidal effect and associated properties of non-electrolytic hydrochloride water on foodborne pathogenic bacteria. This is an interesting topic and a simple methodological aproach to identifying the differences among the SAEW and NEHW against some important foodborne pathogens. However, the Introduction section lacks important information and together with the Materials and methods, the Results are not appropriately written to be published in Foods. Unfortunately, the paper contains few novelties to be published in its present version. I will give some detailed comments to the authors to see if the manuscript can be improved significantly.

Round 2
Reviewer 3 Report
The paper has been improved significantly. Overall, the article is well-structured and appropriately written. But unfortunately, the authors did not reply to a couple of major problems I included in my previous review report. Please, address the comments.
